# Cost-effectiveness of soil-transmitted helminthiasis intervention programmes: A scoping review

Michael Odhiambo[1], Israel Wuresah[2], Geofrey Chepchieng[1], Nikita Kubal[2], Alejandro Krolewiecki[3,4], Charles Mwandawiro[1], Alan Brooks[2], Collins Okoyo[1,5]*

**1** Eastern and Southern Africa Centre of International Parasite Control, Kenya Medical Research Institute, Nairobi, Kenya, **2** Bridges to Development, Seattle, Washington, United States of America, **3** Universidad Nacional de Salta, Instituto de Investigaciones de Enfermedades Tropicales/ CONICET, Orán, Argentina, **4** Innovation, Mundo Sano, Madrid, Spain, **5** Department of Epidemiology, Statistics and Informatics, Kenya Medical Research Institute, Nairobi, Kenya

\* comondi@kemri.go.ke

## Abstract

### Background

Soil-transmitted helminthiases (STH) infect over 1.5 billion people worldwide, causing serious health issues, especially among women of reproductive age and school-age children. Mass drug administration (MDA) using albendazole or mebendazole is the main method of control. However, high reinfection rates and limited integration with water, sanitation, and hygiene (WASH) and routine primary health care programs make the effectiveness of long-term control and elimination challenging.

### Methods

This scoping review aimed to systematically map and summarize cost-effectiveness evidence for STH control and elimination efforts. Following Arksey and O'Malley's framework and Preferred Reporting Items for Systematic reviews and Meta-Analysis extension for Scoping Reviews (PRISMA-ScR) guidelines, we identified 22 studies published from 1993 to 2025 through PubMed and Google Scholar.

### Principal findings

Most studies focused on MDA intervention with school or community-based delivery platforms. School-based delivery costs ranged from $0.03 to $0.76 per child, and community-wide delivery costs ranged from $0.27 to $1.74 per individual. Community-wide programs were estimated to have greater impact and cost-effectiveness in high-prevalence areas, costing $28 to $198 per disability-adjusted life year averted. Programs that delivered multiple disease interventions (integrated programs) showed the highest economic returns, primarily due to shared delivery platforms and reduced operational costs. The main factors influencing cost-effectiveness included treatment

**Data availability statement:** All relevant data are included within the paper.

**Funding:** This study was conducted as part of the Stopping Transmission Of intestinal Parasites (STOP2030) Consortium, funded by the Global Health European and Developing Countries Clinical Trials Partnership 3 (EDCTP3) Joint Undertaking and its members, supported by the European Union's Horizon Europe research and innovation programme, grant number 101103089 awarded to A. K. The funders had no role in study design, data collection and analysis, decision to publish, or preparation of the manuscript.

**Competing interests:** The authors have declared that no competing interests exist.

coverage, baseline prevalence, and delivery costs. Evidence gaps for cost-effectiveness still exist for preschool-aged children and integration with WASH.

## Conclusion

This scoping review showed that context affects the cost-effectiveness of STH intervention programs. Community-wide and integrated MDA strategies offered more economic value than school-based delivery in areas with high prevalence.

---

## Author summary

Soil-transmitted helminthiases (STH) infect more than 1.5 billion people world-wide. These parasites cause anemia, poor nutrition, and impaired growth and learning, particularly among children and women of reproductive age (WRA). The main strategy to control STH has long been mass drug administration (MDA), where medicines such as albendazole and mebendazole are given regularly, usually in schools and communities. While this has had a significant impact on reducing the infections, reinfections are still high, especially in areas with poor water, sanitation, and hygiene (WASH) practices. In this study, 22 published studies on the cost-effectiveness of STH control programs were reviewed. School-based deworming programs cost less than community-wide approaches, but community-wide programs were more cost-effective and achieved a greater reduction in infections in areas with high prevalence. The study also identified important evidence gaps. Few studies included preschool-aged children (Pre-SAC) or WRA, and limited studies assessed the impact of combining MDA with WASH interventions. Overall, the cost-effectiveness of STH control varies by context. Community-wide MDA is more effective in high-prevalence areas, while school-based delivery may be sufficient in lower-prevalence settings depending on the dominant STH species and their transmission dynamics. The critical evidence gap between Pre-SAC and WRA underscores the need to explore the integration of MDA into routine primary care.

## 1. Introduction

Soil-transmitted helminthiasis (STH) remains a significant health concern, particularly in low and middle-income countries (LMICs) [1]. Common STH include roundworm (*Ascaris lumbricoides*), whipworm (*Trichuris trichiura*), and hookworms (*Necator americanus* and *Ancylostoma duodenale*). They are most prevalent in sub-Saharan Africa, South Asia, and parts of Latin America [1,2]. These parasites mainly infect humans through penetration of the skin by infected larvae or ingestion of eggs present in contaminated soil, water, or food [3].

Globally, over 1.5 billion people are infected with these parasites, with women of reproductive age (WRA) and school-age children (SAC) being particularly vulnerable to the

morbidity associated with STH, such as anemia, malnutrition, impaired growth, and learning [2]. While these infections rarely cause death, prolonged infection can lead to serious issues like iron deficiency (anemia), malnutrition, stunted growth, and even impaired cognitive development in children, which can result in lasting educational and economic disadvantages [4,5].

For over a decade, mass drug administration (MDA), using a single dose of albendazole or mebendazole, has been the primary strategy for control of STH. These programs, often carried out in schools, have significantly reduced infection rates [6]. However, reinfection remains a common problem, especially in places without access to clean water, improved sanitation, and hygiene education [7].

This ongoing challenge has sparked global interest in more comprehensive approaches. New strategies for STH control and elimination aim to combine MDA with water, sanitation, and hygiene (WASH) initiatives, along with health education and social behavior change (SBC). Furthermore, in areas where various parasites are prevalent, the use of combined therapies, such as administering ivermectin and albendazole together, shows promise for more comprehensive and successful parasite control [8].

Furthermore, recent evidence from Phase 2/3 clinical trials has demonstrated that the co-formulation of albendazole and ivermectin in a fixed-dose combination (FDC) significantly improves treatment efficacy against *Trichuris trichiura* compared to the standard single-dose albendazole monotherapy. However, we still need to learn more about their economic benefits through further research.

Although the need to improve the integration and sustainability of STH control and elimination efforts is well recognized, there is notably less evidence from economic evaluations of these interventions. Many studies have looked at the epidemiological impact of MDA, WASH, and education strategies, but few have applied cost-effectiveness models to compare these impacts with costs [9–11].

The few existing cost-effectiveness studies further differ in their methods, the range of interventions and delivery platforms, outcome measures, and settings of implementation, which makes it hard to compare them. A systematic mapping of the existing evidence is required to identify what has been modeled, how it has been modeled, and the gaps remaining.

This scoping review aimed to gather information on the various STH interventions, analytical methods, and assumptions used in STH cost-effectiveness modeling. This will help inform future STH modeling studies and strengthen the economic understanding of STH elimination programs.

We specifically aimed to (a) identify and describe the modeling studies that evaluate the cost-effectiveness of STH control and elimination programs, (b) detail the economic evaluation methods used across these studies, (c) summarize the types of interventions assessed and the approach of delivery, including MDA, primary health care (PHC), WASH programs, health education, and integrated programs, and (d) compare the reported economic outcomes and identify the specific factors that influence cost-effectiveness.

## 2. Materials and methods

A review protocol for this study was published in the Open Science Framework on September 8, 2025, with updates on September 28, 2025, and can be accessed at https://osf.io/cz5xj. The writing of the manuscript of this scoping review followed the Preferred Reporting Items for Systematic reviews and Meta-Analysis extension for Scoping Reviews (PRISMA-ScR) guidelines. The PRISMA-ScR checklist can be found in the S1 File (PRISMA-ScR Checklist). An initial search strategy and selection criteria were developed to guide the development of this scoping review. We used the five-step scoping review process that Arksey & O'Malley [12] proposed, which included research question identification, study identification, relevant study selection, data extraction, and result presentation.

### 2.1. Review questions

Given the need to strengthen the economic evidence for STH control and elimination programs, this scoping review mapped and described modeling studies that looked at the cost-effectiveness of STH interventions. The aim was to

indicate the types of interventions studied, the modeling methods used, and the factors that affect the reported economic results. This was intended to inform future modeling efforts and support the economic case for including FDC treatment in STH control and elimination strategies.

## 2.2. Study identification

Study identification search was based on two databases, PubMed and Google Scholar, for published records or studies on cost and cost-effectiveness using a predefined search strategy. We included the Boolean logic operators "AND" (to return results when both keywords are met) and "OR" (to return results when at least one of the keywords is met), together with keywords such as cost, cost-effectiveness, and names of diseases of interest (soil-transmitted helminthiasis, soil-transmitted helminths, STH, *Ascaris*, *Ascaris lumbricoides*, *A. lumbricoides*, hookworm(s), worm(s), roundworms, *Ancylostoma duodenale*, *Necator americanus*, *Trichuris trichura*, *T. trichura*, whipworm(s), *Strongyloides stercoralis, S. stercoralis*, human parasitic disease, and intestinal nematode infections) in our search strategy. An advanced search was conducted in PubMed using the search queries attached as Appendix 1. The queries were then modified to suit the Google Scholar search engine.

## 2.3. Eligibility, screening, and selection criteria

Studies that performed economic analyses, specifically cost-effectiveness models or combined cost and cost-effectiveness analyses, on STH were included. This includes intestinal worms, *Strongyloidiasis,* and other STH infections listed in the search query. Eligible studies were published in English, did not include protocols or reviews, and examined preventive chemotherapy interventions such as albendazole, mebendazole, levamisole, or ivermectin, or other auxiliary interventions like WASH, PHC, health education, SBC, or integrated programs. We placed no restrictions on publication year or geographical location.

All records were imported into Mendeley Reference Manager, which served as the automated tool for identifying and removing duplicate records. Screening took place in two stages. First, we reviewed titles and abstracts to identify studies focused on cost, cost-effectiveness, economic modeling, or related economic evaluations. We excluded records that did not meet the inclusion criteria. Second, four reviewers independently assessed the full texts to confirm each study's relevance to the specified diseases and interventions. We resolved discrepancies through discussion and sought periodic input from senior reviewers. Final inclusion followed the PRISMA-ScR guidelines [13]. Table 1 shows the rationale and criteria for eligibility of including studies in this review.

### 2.3.1. Quality assessment of the extracted records.
Although established critical appraisal tools for economic evaluations exist, such as the Drummond checklist and Joanna Briggs Institute (JBI), in keeping with the objectives of this review, we did not apply these formal tools. Instead, during the full-text review and data extraction process, we took into account descriptive indicators of quality, such as records with accessible cost-effectiveness metrics (e.g., Incremental Cost-Effectiveness Ratios (ICERs), Disability-Adjusted Life Years (DALYs) averted, unambiguous economic evaluation techniques, and relevance to the targeted helminths and the interventions evaluated.

## 2.4. Data extraction and reporting

The data extraction was conducted systematically using a consistent method by four reviewers, following the decision on what information to extract. Disagreements were discussed and resolved among the reviewers with the help of two other subject-level experts. The following details were gathered from the records: author name, publication year, title, location (country or World Health Organization (WHO) region), target group/population, target helminth, methods, outcome metrics, transmission parameters, cost-effectiveness parameters, intervention, comparison group, data types, and key results. This information was then checked to ensure its accuracy and consistency across each record. We entered the extracted information into a template we created in Microsoft Excel for descriptive analysis.

**Table 1. Eligibility criteria for publications in the review and the rationales for selecting these criteria.**

| Category | Criterion (Inclusion: white, Exclusion: orange | Rationale |
|---|---|---|
| **Population** | Studies conducted on human populations (all ages, including children, adults, women of reproductive age, and communities). | The review is concerned with the cost-effectiveness of interventions targeting STH infections among humans. |
| | Studies conducted on non-human populations (e.g., animal-only studies, lab-only models). | This review is not concerned with studies targeting non-human populations. |
| **Concept** | Studies that perform economic analyses, specifically cost-effectiveness, or that combine cost and cost-effectiveness analyses of STH interventions. | This review targeted studies that performed economic evaluations of STH interventions to guide policy and program design. |
| | Studies not addressing STH; studies without cost, cost-effectiveness, or economic evaluation metrics; studies only epidemiological or clinical without economic modeling. | This review aims to map economic evidence for soil-transmitted helminthiasis interventions. Studies not addressing STH are outside the disease scope. |
| **Context** | Global scope; no restriction on country or region. | STH is a global health problem, so all geographical regions were considered relevant. |
| **Interventions** | Preventive chemotherapy (albendazole, mebendazole, levamisole, ivermectin); auxiliary interventions (WASH, PHC, health education, social behavior change, integrated programs). | Review targeted both pharmaceutical and integrated interventions relevant to STH. |
| | Studies without preventive chemotherapy or auxiliary interventions related to STH control/elimination. | This review focused specifically on interventions with direct or indirect impact on STH control/elimination. |
| **Study Design / Evidence Sources** | Primary studies (modeling studies, field studies, economic evaluations) published in English; peer-reviewed. | Ensured inclusion of studies with extractable cost-effectiveness data, excluding secondary or inaccessible sources. |
| | Protocols, reviews, commentaries, non-English studies, incomplete or restricted access papers. | These do not provide completed economic analyses or results. |

**2.4.1. Outcomes and prioritization.** The primary outcomes extracted from the included studies were economic indicators, specifically ICERs, DALYs averted, and cost per infection or morbidity reduction. These outcomes were prioritized due to their critical relevance for guiding resource allocation and informing public health policy decisions, particularly in settings with constrained resources and persistent STH transmission. Additionally, this scoping review aimed to identify key parameters, assumptions, and methodological approaches commonly used in existing economic evaluations. Mapping these elements is essential for the future development of a comprehensive and robust cost-effectiveness model. Such a model would explicitly evaluate the economic feasibility of novel interventions, including simplified FDC therapies, and assess their suitability for broader target populations such as preschool-aged children (Pre-SAC) and adults. The included studies utilized a variety of data inputs, ranging from primary field-measured costs (actual) to secondary estimated parameters (assumed).

## 2.5. Data analysis

The analysis followed Arksey & O'Malley [12] framework for gathering, summarizing, and reporting scoping review results. We extracted relevant information from the final set of records and analyzed it descriptively to look at its nature, distribution, and scope. We visualized the findings in tables and graphs that show the geographic distribution of studies, types of interventions, target populations, target helminths, and methods of analysis. We grouped outcomes into common themes, patterns, and trends to synthesize the results. All analyses were done in Microsoft Office Excel version 16.100.3.

# 3. Results

## 3.1. Description of study selection

Drawing on the Arksey and O'Malley framework, we synthesized findings from twenty-two studies, mapping their study characteristics, methods, and outcomes, while also identifying critical gaps in the evidence base. A PRISMA flow diagram (Fig 1) summarizes the stages of the study selection process, from identification through screening and final inclusion. The included studies represented a broad geographical distribution, with the majority of the studies conducted in sub-Saharan Africa, followed by East and Southeast Asia, and a smaller number from Latin America. Two studies were modeling-based and not tied to specific field sites.

## 3.2. Target populations, geographic scope, and intervention modalities

Table 2 outlines the details extracted from each study, which includes: the author, publication year, geographic location, primary research focus, target population, type of intervention assessed, and key economic outcomes reported.

### 3.2.1. Geographical distribution.
Across the 22 studies, sub-Saharan Africa accounted for the largest share of evidence (59%, n = 13), with particularly high representation from Kenya (n = 3) and Côte d'Ivoire (n = 3), followed by Uganda (n = 2), Nigeria (n = 2), Madagascar (n = 1), Niger (n = 1), and Ghana (n = 1). Southeast/East Asia contributed 18% (n = 4), including China, Vietnam, the Philippines, and Lao PDR. Latin America was represented by a single study from Brazil (5%). Fig 2 presents a geographical map showing the countries represented in the included studies, while noting that two studies were excluded from the figure because they were modeling-based and not tied to a specific country.

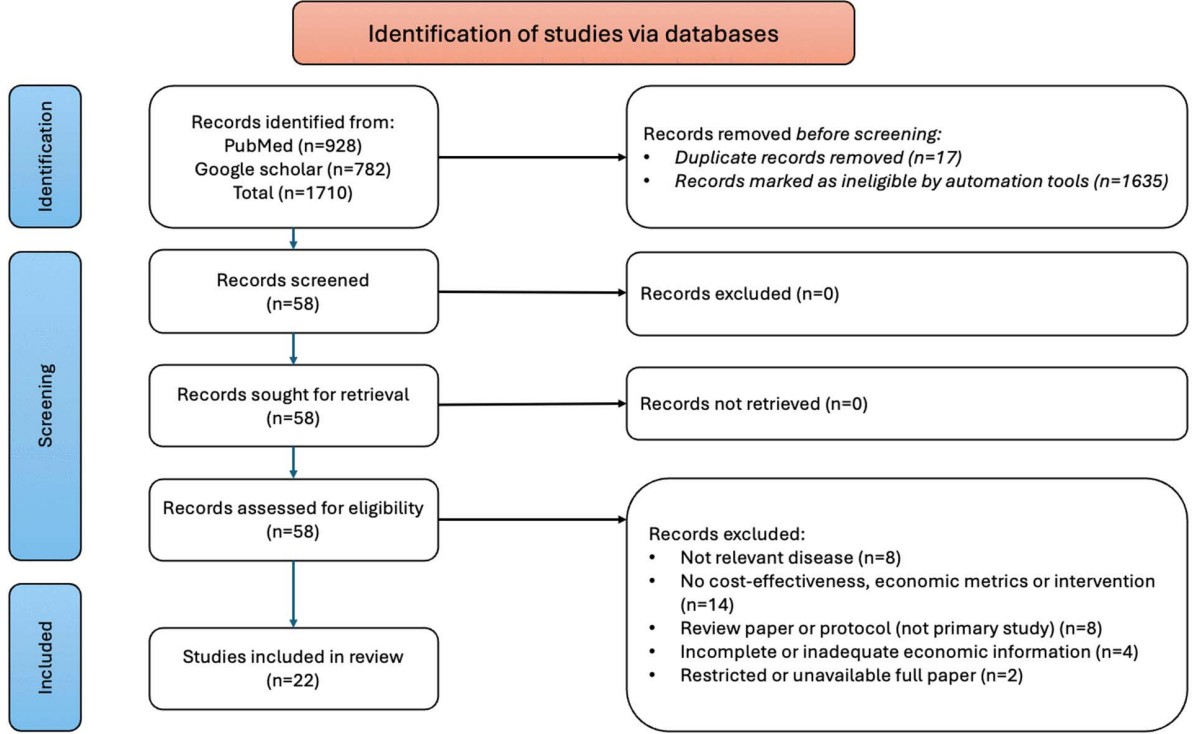

**Fig 1. PRISMA flow diagram showing the identification, screening, and inclusion processes.**

**Table 2. Summary characteristics of studies included in the scoping review.**

| Author, year | Location (Country/Region) | Focus | Population | Intervention | Economic Outcome |
|---|---|---|---|---|---|
| Zhu et al., 2025 [28] | China (Western Pacific) | System dynamics modelling for hookworm in China | Whole population | WPD[1], KPD[2], ED[3], EVD[4] + health education | ICER[5] per infection reduced |
| Minnery et al., 2024 [29] | Kenya (Africa) | Survey cost-effectiveness for Kenyan deworming | Schoolchildren | Geostatistical vs cluster survey | ICER per survey precision |
| Coffeng et al., 2024 [19] | Not country-specific | MDA ivermectin for strongyloidiasis | Children 5–15 and community 5 + years | Annual ivermectin MDA | Cost/DALY[6] averted |
| Coffeng et al., 2023 [30] | Ethiopia and Tanzania (Africa) | Diagnostic cost-effectiveness framework | Schoolchildren | Diagnostic tools cost-effectiveness | Cost per reliable diagnostic result |
| Trinos et al., 2023 [14] | Vietnam and Philippines (Western Pacific) | MDA for hookworm in Vietnam and Philippines | Entire community including preschool aged children | MDA with Albendazole | ICER per DALY and infection-years averted |
| Cha et al., 2019 [31] | Sudan (Africa) | Nationwide STH and schistosomiasis survey in Sudan | School children | Targeted ecological MDA | Efficiency of ecological vs district-level MDA |
| Pullan et al., 2019 [7] | Kenya (Africa) | Community-wide vs school-based MDA in Kenya | Children 2–14 and community 2 + years | Annual vs biannual MDA | Cost/person and prevalence reduction |
| De Neve et al., 2018 [21] | Madagascar (Africa) | Integrated PC in Madagascar | School-aged children | School-based PC[7], integrated NTDs | ICER per DALY, benefit-cost ratios |
| Okello et al., 2018 [22] | Northern Lao PDR (Western Pacific) | One Health integration: *T. solium* + STH + swine fever | Communities & pigs | Integrated One Health MDA & pig vaccination | Cost/DALY and zDALY[8] averted |
| Thakur et al., 2018 [24] | Ghana (Africa) | Optimizing MDA by age/risk in Ghana | Children & adults (risk stratified) | MDA by age/risk mix | Cost/infection reduction per strategy |
| Lo et al., 2016 [25] | Côte d'Ivoire (Africa) | Global MDA guideline assessment | Children, preschool, adults | Annual/biannual PC, integrated | ICER per DALY, prevalence thresholds |
| Bartsch et al., 2016 [23] | Brazil (Latin America) | Hookworm vaccine + MDA in Brazil | Children & adults | MDA vs vaccine | ICER for vaccine & MDA |
| Turner, et al., 2016 [26] | Uganda (Africa) | Economies of scale in Uganda MDA | Schoolchildren | Scaling MDA with economies | Cost-effectiveness under economies of scale |
| Turner, Truscott, & Anderson, 2016 [17] | Côte d'Ivoire (Africa) | Community-wide vs school-only MDA in Côte d'Ivoire | Entire community | Community-wide MDA vs school | ICER per DALY averted |
| Lo et al., 2015 [15] | Côte d'Ivoire (Africa) | Community-wide integrated PC in Côte d'Ivoire | Community-wide, all ages | Integrated community-wide PC | ICER per DALY averted robust to high costs |
| Evans et al., 2011 [20] | Nigeria (Africa) | Triple-drug MDA Nigeria | School-aged children | Triple-drug MDA (ivermectin, albendazole, and praziquantel) | Cost savings per treatment delivered |
| Leslie et al., 2011 [32] | Niger (Africa) | School vs community MDA in Niger | Children & adults | School vs community MDA | Cost/infection averted, children vs adults |
| Hall et al., 2009 [33] | Africa, Asia, Latin America | MDA thresholds cost-effectiveness | Preschool & school-aged children | WHO vs 3-tier thresholds | Cost/diseased person treated |
| Brooker et al., 2008 [18] | Uganda (Africa) | Nationwide school-based MDA in Uganda | Schoolchildren | Nationwide school MDA | ICER per anemia averted |
| Guyatt, 2003 [34] | Kenya (Africa) | Program delivery cost in Kenya | Schoolchildren & adults | School vs community delivery | Cost per treatment, affordability |
| Holland et al., 1996 [35] | Nigeria (Africa) | Levamisole MDA in Nigeria | Community (children focus) | Mass, targeted, selective MDA | Cost per 1,000 epg reduced |
| Guyatt et al., 1993 [27] | Model-based (not location-specific) | Dynamic CEA Ascaris treatment frequency | Community-wide | Annual, biannual MDA | Cost per case averted by frequency |

[1]Whole population deworming [2]Key population deworming [3]Examination-and-deworming [4]Examination and voluntary deworming [5]Incremental cost-effectiveness ratio [6]Disability-adjusted life years [7]Preventive chemotherapy [8]Zoonotic Disability-adjusted life years.

**PLOS** ✹ Neglected Tropical
Diseases

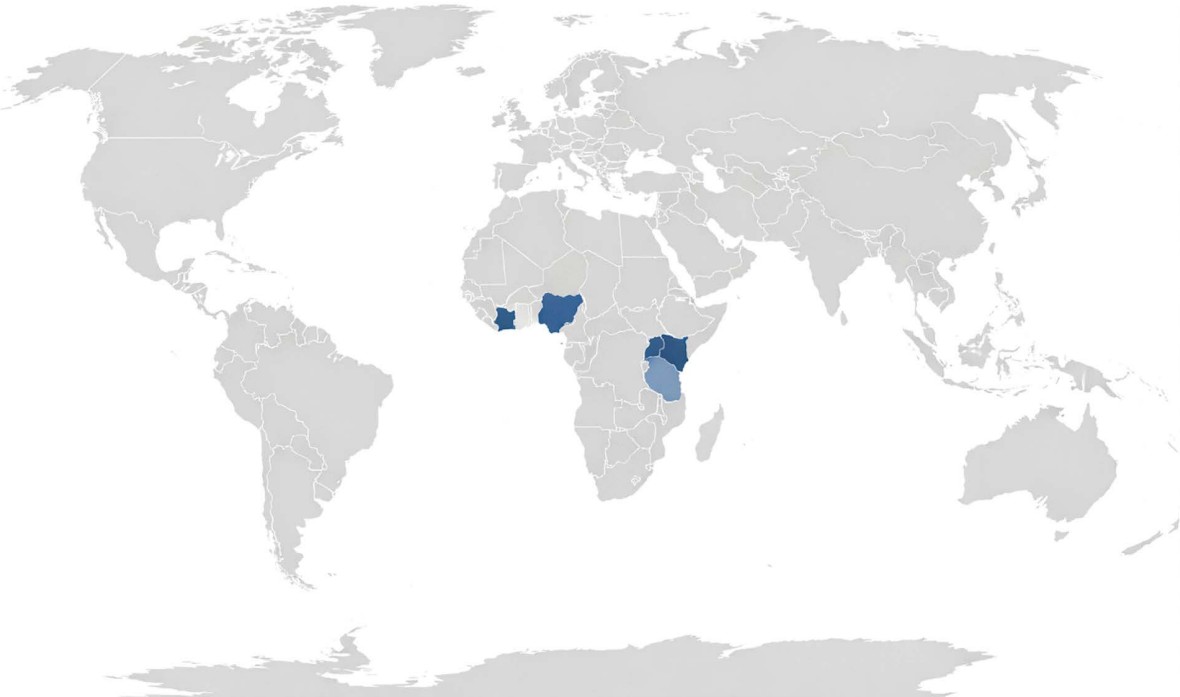

**Fig 2. Geographic distribution of included studies.** Countries are shaded according to the number of studies conducted in each location. The maps were generated using QGIS v3.42 (www.qgis.org; vector map: Natural Earth: www.naturalearthdata.com/about/terms-of-use). The basemap used is the Natural Earth (public domain).

**3.2.2. Target populations.** Most programs focused on school-aged children (41%, n = 9) (Fig 3), reflecting WHO guidelines. A larger proportion (59%, n = 13) expanded to entire communities, including adults, WRA, and in some cases, Pre-SAC, with evidence that this expansion improves cost-effectiveness in moderate to high prevalence settings [7,14,15].

**3.2.3. Types of intervention.** The majority of studies (68%, n = 15) examined MDA, either school-based, community-wide, or integrated into neglected tropical disease (NTD) programs. School-based MDA programs were delivered at low cost, ranging from $0.03 to $0.76 per child treated, and were often integrated with other NTD control activities to improve efficiency (Brooker et al., 2008; De Neve et al., 2018; Leslie et al., 2011). Evidence from Uganda and other settings demonstrated that economies of scale (the cost advantage that arises with increased output, where the average cost per unit decreases as the number of people treated increases [16]) significantly reduced per-child costs as coverage expanded [17,18].

Community-wide MDA interventions were frequently more cost-effective per DALY averted than school-based approaches, particularly in high-prevalence settings with substantial adult infection burdens. Reported ICERs for these programs ranged from $28.55 to $198 per DALY averted [19,14]. Expanding treatment beyond SAC increased the overall impact on transmission and improved cost-effectiveness.

Integrated or multi-disease delivery models also showed strong economic performance. In Nigeria, bundling ivermectin, albendazole, and praziquantel into a single delivery platform reduced per-person delivery costs by 41% [20]. Favorable DALY and benefit-cost ratios were also achieved in Lao PDR and Madagascar [21,22]. One study modeled the combination of a hypothetical hookworm vaccine with MDA in Brazil, aiming to reduce reinfection rates and extend deworming benefits [23].

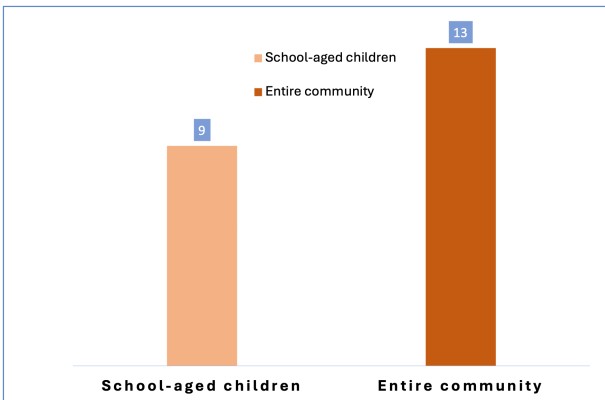

**Fig 3. Distribution of target populations among included studies.**

## 3.3. Economic outcomes and methodological approaches

**3.3.1. Economic metrics.** Cost per DALY averted was the most common outcome, reported in 45% of studies, with values ranging from US$14 for a One Health program in Lao PDR to US$198 for school-based preventive chemotherapy targeting strongyloidiasis. Incremental cost-effectiveness ratios were widely used; for example, community-wide versus school-only MDA in Côte d'Ivoire yielded ICERs between US$127 and US$167 per DALY averted [17,15].

Cost per treatment delivered varied substantially by delivery mode. School-based programs ranged from US$0.03 to US$0.76, while community-based programs ranged from US$0.27 to US$1.74 per treatment. In community-wide contexts, the cost per infection-year averted was often under US$1 [14]. Benefit-cost ratios reached as high as US$31 for integrated school-based programs [21].

**3.3.2. Methodological diversity.** Studies applied diverse modeling approaches. Dynamic transmission models were used in 32% of the studies (n = 7) [24,25,26,15,27,23], supporting expansion of MDA to Pre-SAC and adults, even under higher assumed delivery costs [15]. Program delivery cost analyses, accounted for 32% of the studies (n = 7), highlighted affordability drivers and economies of scale. Other frameworks included Markov models (5%, n = 1), system dynamics models (5%, n = 1), and stochastic individual-based simulations (5%, n = 1), which confirmed robustness across assumptions for cure rates, delivery costs, and time horizons [19].

## 3.4. Cross-cutting findings

**3.4.1. Recurring patterns.** Community-wide MDA consistently outperformed school-only strategies in DALYs and infection-years averted. Integrated platforms addressing multiple diseases produced the highest returns per dollar spent. Programs benefited from economies of scale, with per-person costs declining as coverage expanded.

**3.4.2. Key drivers of cost-effectiveness.** Sensitivity analyses were reported in 9 of 22 studies (44%), most commonly testing the impact of drug delivery costs, baseline prevalence levels, and treatment coverage on cost-effectiveness outcomes. For example, Lo et al. showed that community-wide treatment strategies remained highly cost-effective when adult delivery costs were assumed to be ten times higher than those for children [15]. Coffeng et al. [19] demonstrated that community-wide ivermectin MDA consistently remained the preferred strategy under a wide range of assumptions for cure rates, delivery costs, time horizons, and coverage. Similarly, Trinos et al. used probabilistic sensitivity analysis to confirm the robustness of community-wide MDA in Vietnam and the Philippines. Other studies, including De Neve et al. [21], Okello et al. [22], Bartsch et al. [23], and Brooker et al. [18] incorporated parameter uncertainty or scenario analyses, highlighting the role of program costs, drug efficacy, and prevalence thresholds as

key drivers. Collectively, these analyses strengthen confidence in the conclusion that community-wide and integrated approaches tend to outperform school-based MDA, though the lack of sensitivity analyses in over half of the studies limits cross-study comparability and generalizability.

### 3.4.3. Gaps in the evidence.
Geographic gaps remain, with limited evidence from the Western Pacific, the Caribbean, and most of Latin America. Few studies included preschool-aged children, and none considered children under two years. Analyses of fixed-dose combination therapies are scarce, likely due to their novelty. Evidence on long-term sustainability, integration with WASH, and broader One Health approaches is also limited. Diagnostic and surveillance-focused evaluations rarely reported disability-adjusted based outcomes, reducing comparability with treatment-focused analyses. Also, only one study specifically addressed *Strongyloides stercoralis* [19].

## 4. Discussion

This scoping review revealed that community-wide and integrated MDA approaches tend to be more cost-effective than school-based delivery in high-prevalence settings, particularly when combined with other NTD programs and scaled to larger populations.

The ICER reported by the studies ranged from US$14 to US$198 per DALY averted. The variation was largely due to differences in coverage levels, integration, and baseline prevalence. That is, the cost-effectiveness of STH interventions is highly context-specific, influenced by delivery strategy, population coverage, and program integration. In high-prevalence settings, community-wide MDA was more impactful than school-based delivery because adult treatment interrupts transmission, but reinfection limited overall effects [14,15]. However, the operational feasibility of such approaches relies on the ability to reach geographically reduced infected adult populations and health systems' willingness to incur higher upfront costs. Integrated NTD campaigns consistently improved cost-effectiveness by sharing delivery platforms and reducing per-person costs [22,20].

Further, we found uneven amounts of data on cost-effectiveness from endemic regions. Most studies focused on Sub-Saharan Africa, with limited evidence emerging from other endemic regions like Latin America, the Western Pacific, or the Caribbean. These gaps are particularly consequential because economic efficiency does not solely rely on biological impact. It also incorporates local delivery costs, local health system capacity, baseline prevalence patterns, and other factors [7]. With a limited geographical scope, policies are likely to be centered on strategies tailored to scenarios with mismatched epidemiological or economic attributes.

Additionally, differences in methodological approaches were noticed across the studies included in this review, which makes comparability difficult. Expanded coverage scenarios were often associated with more favorable cost-effectiveness ratios in dynamic transmission models, while static cost analyses were more centered on near-term affordability and gave a narrower view. Inconsistent parameter assumptions, especially concerning cure rates, delivery costs, and time frames, make cross-study synthesis challenging as well. The interpretability and relevance of policies derived from cost-effectiveness studies could be improved with more uniform modeling approaches combined with clear sensitivity analyses [19].

One major gap we found in our review is the lack of evidence and intervention among important at-risk groups. Pre-SAC and WRA are, in general, underrepresented in economic evaluations, and cost-effectiveness analyses conducted to date also seem to have completely ignored these critical groups. This gap is concerning in that these groups are especially susceptible to morbidity of STH infections, including stunted growth and development (among Pre-SAC), and anemia [4]. In the absence of economic evidence for these children, policymakers are less able to allocate resources. This makes it more difficult to design targeted interventions for some of the most at-risk groups, ultimately undermining efforts to eliminate STH at the population level.

A key gap in the literature is the limited examination of the long-term sustainability and cost-effectiveness of interventions implemented without concurrent WASH improvements. Many studies evaluate the immediate outcomes of MDA but fail to assess whether the benefits persist, particularly if reinfection rates return to baseline once treatment stops.

Sustainable reduction of STH infections requires not only consistent access to medication but also significant environmental improvements that lower the risk of reinfection. The failure to evaluate the combination of WASH with MDA in the long term suggests that these strategies could be only temporally effective. The variability in context and delivery costs across countries was also not consistently accounted for in the included studies. This limits the comparability of findings across settings and highlights the need for future analyses to explicitly incorporate local economic and health system factors when modeling cost-effectiveness.

This scoping review is particularly important for assessing the possible cost-effectiveness of a new FDC of albendazole and ivermectin for STH. Integrated MDA campaigns support the claim that treatments, when combined and delivered in a single delivery event, could significantly reduce cost per person and increase the return on investment. This is particularly true in areas where STH epidemiology calls for both drugs to be used [8]. The FDC's broader activity could achieve a greater health impact by concurrently eliminating STH and other parasites, thus increasing the DALYs averted per treatment round. Nonetheless, without empirical cost-effectiveness estimates for FDC, modeling will need to account for potential gains in efficiency against uncertainties in manufacturing cost, procurement pricing, and adoption feasibility. The literature suggests that such an intervention would likely be most cost-effective in high-prevalence, co-endemic settings where community-wide coverage is feasible and integration with existing NTD programs can be achieved.

From the reviewed studies, there was a notable absence of costs associated with monitoring and evaluation (M&E) frameworks in the cost-effectiveness analyses. Robust M&E systems are essential to track program coverage, impact, treatment compliance, reinfection dynamics, and the sustainability of the interventions. Integrating M&E costs into cost-effectiveness models would complete the intervention cycle and enable more accurate policy guidance and resource allocation.

Nevertheless, this review has some notable limitations. First, the reliance on English-language publications may have excluded relevant evidence from non-English sources, notably literature from the Americas published in Spanish. Second, the heterogeneity of economic metrics, modeling assumptions, and intervention designs across studies limited our ability to perform direct quantitative comparisons. Finally, given the inclusion of studies spanning more than three decades, differences in global and local economic conditions, inflation, drug pricing, and healthcare infrastructure could influence the comparability of cost-effectiveness results. As a scoping review, we did not perform temporal standardization or inflation adjustments. Therefore, findings should be interpreted as relative rather than absolute economic measures.

In conclusion, this scoping review demonstrates that context has an impact on the cost-effectiveness of STH interventions, with community-wide and integrated MDA strategies typically providing higher economic value than school-based delivery in high-prevalence settings. Improved efficiency was consistently driven by utilizing economies of scale and integrating with other NTD control activities. However, there is uncertainty regarding the generalizability of the findings of this study to other endemic regions due to the fact that the evidence base of the included studies was heavily concentrated in sub-Saharan Africa. Cross-study comparability is further restricted by methodological diversity and inconsistent parameter assumptions. The review also identifies important evidence gaps for interventions that combine environmental and pharmaceutical approaches, as well as for high-risk but underrepresented groups, such as Pre-SAC and WRA.

The findings carry direct relevance for assessing the potential cost-effectiveness of the FDC of albendazole and ivermectin. Experience from integrated MDA campaigns suggests that such a formulation could improve efficiency and impact, particularly in areas with multiple STH burden and high-prevalence contexts where community-wide coverage is achievable. Nonetheless, empirical economic evaluations of FDC for STH control are lacking, highlighting the need for robust modeling and field-based studies to confirm these projections.

## Supporting information

**S1 File. PRISMA-ScR Checklist v2.0.** Preferred Reporting Items for Systematic reviews and Meta-Analyses extension for Scoping Reviews checklist, detailing the location of reported items within the manuscript. *From:* Page MJ, McKenzie JE, Bossuyt PM, Boutron I, Hoffmann TC, Mulrow CD, et al. The PRISMA 2020 statement: an updated guideline for

reporting systematic reviews. BMJ 2021;372:n71. https://doi.org/10.1136/bmj.n71. This work is licensed under CC BY 4.0. To view a copy of this license, visit https://creativecommons.org/licenses/by/4.0/.
(DOCX)

**S1 Data. Search queries used in the PubMed database.**
(DOCX)

## Author contributions

**Conceptualization:** Alan Brooks, Collins Okoyo.

**Data curation:** Michael Odhiambo, Israel Wuresah, Geofrey Chepchieng, Nikita Kubal, Collins Okoyo.

**Formal analysis:** Michael Odhiambo, Israel Wuresah, Geofrey Chepchieng, Nikita Kubal, Collins Okoyo.

**Funding acquisition:** Alejandro Krolewiecki, Charles Mwandawiro, Collins Okoyo.

**Investigation:** Alejandro Krolewiecki, Charles Mwandawiro, Alan Brooks, Collins Okoyo.

**Methodology:** Michael Odhiambo, Israel Wuresah, Geofrey Chepchieng, Nikita Kubal, Alan Brooks, Collins Okoyo.

**Project administration:** Alejandro Krolewiecki, Charles Mwandawiro, Alan Brooks, Collins Okoyo.

**Resources:** Alejandro Krolewiecki, Alan Brooks, Collins Okoyo.

**Software:** Michael Odhiambo, Israel Wuresah, Geofrey Chepchieng, Nikita Kubal, Collins Okoyo.

**Supervision:** Michael Odhiambo, Israel Wuresah, Geofrey Chepchieng, Nikita Kubal, Alejandro Krolewiecki, Alan Brooks, Collins Okoyo.

**Validation:** Alejandro Krolewiecki, Alan Brooks, Collins Okoyo.

**Visualization:** Michael Odhiambo, Israel Wuresah, Geofrey Chepchieng, Nikita Kubal, Collins Okoyo.

**Writing – original draft:** Michael Odhiambo, Israel Wuresah, Geofrey Chepchieng, Nikita Kubal, Alejandro Krolewiecki, Charles Mwandawiro, Alan Brooks, Collins Okoyo.

**Writing – review & editing:** Michael Odhiambo, Israel Wuresah, Geofrey Chepchieng, Nikita Kubal, Alejandro Krolewiecki, Charles Mwandawiro, Alan Brooks, Collins Okoyo.

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
