## [Decision Letter · Decision Letter 0]

22 Jan 2026

PNTD-D-25-01829

Cost-effectiveness of soil-transmitted helminthiasis intervention programmes: A scoping review

Dear Dr. Okoyo,

Thank you for submitting your manuscript to PLOS Neglected Tropical Diseases. After careful consideration, we feel that it has merit but does not fully meet PLOS Neglected Tropical Diseases's publication criteria as it currently stands. Therefore, we invite you to submit a revised version of the manuscript that addresses the points raised during the review process.

Please submit your revised manuscript within by Mar 23 2026 11:59PM. If you will need more time than this to complete your revisions, please reply to this message or contact the journal office at plosntds@plos.org. Please include the following items when submitting your revised manuscript:

We look forward to receiving your revised manuscript.

Kind regards,

Luis Marcos, MD, MPH, FIDSA

Guest Editor

Krystyna Cwiklinski

Section Editor

Shaden Kamhawi

co-Editor-in-Chief

Paul Brindley

co-Editor-in-Chief

**Additional Editor Comments:**

Reviewers comments should be taken into consideration, a new version will be reviewed.

**Journal Requirements:**

1) Please upload all main figures as separate Figure files in .tif or .eps format. For more information about how to convert and format your figure files please see our guidelines:

2) We have noticed that you have uploaded Supporting Information files, but you have not included a list of legends. Please add a full list of legends for your Supporting Information files after the references list.

3) We notice that your supplementary information is included in the manuscript file. Please remove them and upload them with the file type 'Supporting Information'. Please ensure that each Supporting Information file has a legend listed in the manuscript after the references list.

Potential Copyright Issues:

- Figure 2. Please (a) provide a direct link to the base layer of the map (i.e., the country or region border shape) and ensure this is also included in the figure legend; and (b) provide a link to the terms of use / license information for the base layer image or shapefile. We cannot publish proprietary or copyrighted maps (e.g. Google Maps, Mapquest) and the terms of use for your map base layer must be compatible with our CC BY 4.0 license.

**Reviewers' Comments:**

Reviewer's Responses to Questions

**Key Review Criteria Required for Acceptance?**

**Methods**

-Are the objectives of the study clearly articulated with a clear testable hypothesis stated?

-Is the study design appropriate to address the stated objectives?

-Is the population clearly described and appropriate for the hypothesis being tested?

-Is the sample size sufficient to ensure adequate power to address the hypothesis being tested?

-Were correct statistical analysis used to support conclusions?

-Are there concerns about ethical or regulatory requirements being met?

Reviewer #1: Overall, this is a well written and clear paper. The results are compelling, despite the acknowledged limitations. My main concern or question regarding the methodology is whether the studies that were included based their cost-effectiveness analyses on actual (i.e., measured) costs/benefits or on assumed or estimated ones? Is there a way for the authors to ascertain this for each study and to add it to the paper? For example, the single Brazil study seems to have included inputs from several other studies (even some done in Africa).

Minor:

1. Figure 1 describes records marked as ineligible by "automation tools" but these tools are not described in the Methods section. Please clarify.

Reviewer #2: the objective is clear with correct statistical analysis

Reviewer #3: The research question must be well-defined and aligned with the objectives. There is no consistency between the conclusion and the questions posed at the beginning.

Although they chose a scoping review approach, when discussing the economic models used to evaluate these studies, these models are not ultimately considered or mentioned.

The title does not accurately reflect the paper's actual purpose, and, most importantly, it fails to specify the geographical areas studied. I would suggest narrowing the scope to Sub-Saharan Africa.

**Results**

-Does the analysis presented match the analysis plan?

-Are the results clearly and completely presented?

-Are the figures (Tables, Images) of sufficient quality for clarity?

Reviewer #1: 1. Line 262: please clarify that the "hookworm vaccine" is a hypothetical one, since there currently is no approved vaccine for this STH.

2. Table 2 lists the year of publication for each study, but that doesn't necessarily reflect the time period when the data were collected, since publication could have occurred years later. Would it be possible to include the dates that the studies were conducted or during which the data were collected?

Reviewer #2: analysis presented match the analysis plan

Reviewer #3: The results should be aligned with the studies. Table 1 should be improved, as it is currently redundant, and I would recommend eliminating the third column.

The map presented should have two parts: zones showing areas according to prevalence (high, medium, or low), and, either on the same or a separate map, the number of studies in choropleth maps. If the studies only considered English-language literature, I would remove the data concerning Latin America from the study to avoid drawing erroneous conclusions.

**Conclusions**

-Are the conclusions supported by the data presented?

-Are the limitations of analysis clearly described?

-Do the authors discuss how these data can be helpful to advance our understanding of the topic under study?

-Is public health relevance addressed?

Reviewer #1: Overall, the conclusions appear to be supported by the data presented. Limitations are clearly stated.

Reviewer #2: the conclusion supported by data presented, the authors discuss how data can help understanding of strategies for the control of STD.

Reviewer #3: The study draws conclusions based on unclear information, such as defining that population-level interventions are more cost-effective than school-level interventions, when Table 1 only identifies two or three studies with those characteristics.

The limitations do not mention that the cost-effectiveness analyses that should have been used were not performed, and I believe this should be clearly stated in the limitations.

As the authors themselves mention, there are significant biases in the collection and analysis of the data. I suggest improving the analysis, the formulation of the research question, and ensuring consistency throughout the article.

**Editorial and Data Presentation Modifications?**

Reviewer #1: 1. Lines 91-93: how does coformulation of albendazole and ivermectin in a fixed-dose combination provide evidence of effectiveness? I think that what the authors mean to say is that recent evidence from a Phase 2/3 trial (published in TLID) showed improved efficacy of the FDC vs single-dose ALB against T. trichiura. However, the sentence as currently worded does not state this and should be revised for clarity.

2. Line 166: should this be "ratio" instead of "ration" (ICER definition)?

Reviewer #2: These data could have been very useful in general. Because of the limitation to English litterature its sound that in area where STD and others parasite like Loa loa co exist. strategies may be different. I will suggest that the authors expand discussion on this particular situation ( IVM= ivermetin, may be use with caution in loiasis endemic countries ....) where economic benefits of combined ALB/IVM needs further studies.

Reviewer #3: (No Response)

**Summary and General Comments**

Reviewer #1: (No Response)

Reviewer #2: This study map the area covered by mass drug administration for STD control in the world. Its show economic value of community based control over school chidren based control. its demonstrate that community based control is more economical than school children based strategy. We suggest to the authors to expand their discussion in area where STD co exist whit filaria Loa loa

Reviewer #3: I believe that while the article addresses an important topic such as the cost-effectiveness of programs, it fails to integrate the necessary methods for conducting such analyses through a systematic review. I suggest that the title be revised, clearly stating that it is merely a description and not an analysis, as is the conclusion reached.

PLOS authors have the option to publish the peer review history of their article (what does this mean?). If published, this will include your full peer review and any attached files.

Reviewer #1: No

Reviewer #2: No

Reviewer #3: No

**Figure resubmission:** While revising your submission, we strongly recommend that you use PLOS’s NAAS tool (https://ngplosjournals.pagemajik.ai/artanalysis) to test your figure files. NAAS can convert your figure files to the TIFF file type and meet basic requirements (such as print size, resolution), or provide you with a report on issues that do not meet our requirements and that NAAS cannot fix.
---

## [Editor Report · Decision Letter 1]

29 Apr 2026

Dear Dr Okoyo,

We are pleased to inform you that your manuscript 'Cost-effectiveness of soil-transmitted helminthiasis intervention programmes: A scoping review' has been provisionally accepted for publication in PLOS Neglected Tropical Diseases.

Best regards,

Luis Marcos, MD, MPH, FIDSA

Guest Editor

Krystyna Cwiklinski

Section Editor

Shaden Kamhawi

co-Editor-in-Chief

Paul Brindley

co-Editor-in-Chief

Authors have addressed properly all feedback provided by reviewers. Article sounds now more scientifically correct.
